# An Optimized and Standardized Rapid Flow Cytometry Functional Method for Heparin-Induced Thrombocytopenia

**DOI:** 10.3390/biomedicines9030296

**Published:** 2021-03-13

**Authors:** Anne Runser, Caroline Schaning, Frédéric Allemand, Jean Amiral

**Affiliations:** 1Emosis SAS, 11 Rue de l’Industrie, 67400 Illkirch, France; anne.runser@emosis-diagnostics.com (A.R.); caroline.schaning@emosis-diagnostics.com (C.S.); 2SH-Consulting, 68 Quai de l’Oise, 78570 Andrésy, France; jean.amiral@scientific-hemostasis.com

**Keywords:** heparin-induced thrombocytopenia diagnosis, HIT functional assay, flow cytometry

## Abstract

Heparin-induced thrombocytopenia (HIT) is a thrombocytopenia caused by heparin and mediated by an atypical immune mechanism leading to a paradoxical high thrombotic risk, associated with severe morbidity or death. The diagnosis of HIT combines a clinical scoring of pretest probability and laboratory testing. First-line routine tests are antigen binding assays detecting specific antibodies. The most sensitive of these tests have a high HIT-negative predictive value enabling HIT diagnosis to be ruled out when negative. However, HIT-positive predictive value is low, and a functional assay evaluating the pathogenicity of the antibodies should be performed to exclude false-positive results. In contrast to screening assays, functional assays are highly specific but technically challenging, and are thus performed in referral laboratories, where platelet activation is detected using radioactive serotonin (serotonin release assay, SRA) or visually (heparin-induced platelet activation, HIPA). Flow cytometry is a possible alternative. It is, however, currently not widely used, mostly because of the lack of standardization of the published assays. This article describes and discusses the standardization of a HIT flow cytometry assay (HIT-FCA) method, which subsequently led to the development and commercialization of a CE-marked assay (HIT Confirm^®^, Emosis, France) as a suitable rapid HIT functional test.

## 1. Introduction

Heparin is one of the oldest drugs in the pharmacopeia of modern medicine. Heparin is widely used as injectable anticoagulant, with many clinical indications, both for prophylactic and therapeutic purposes. Chemically, heparin is a chain of highly negatively charged glycosaminoglycans, and a variety of heparins were developed that differ by the length of that chain and by their molecular weight, ranging from 6 to 30 kDa for unfractionated forms of heparin (UFH) and from 3 to 6 kDa for low molecular weight heparin (LMWH). In the United States, almost one third of the inpatient population receives heparins during hospitalization (i.e., about 12 million patients per year). LMWH, which does not require intravenous infusion or biological monitoring, has largely replaced UFH as a frontline therapy. However, UFH continues to be broadly used owing to its relative efficacy and safety, low cost, and familiarity to clinicians, especially among selected settings (such as intensive care units or cardiovascular surgery) because of its brief half-life, reversibility with protamine sulfate, and safety in the setting of renal insufficiency.

Although an antithrombotic drug, heparin may trigger an early onset and transient proaggregant effect associated with a mild thrombocytopenia in some patients. This non-immune-mediated effect that does not necessarily require the discontinuation of heparin is called heparin-induced thrombocytopenia type 1, also known as heparin-associated thrombocytopenia. Unfortunately, a small subset of patients treated with heparin will suffer from another and much more severe complication of heparin treatment called heparin-induced thrombocytopenia type 2 (HIT). In HIT, patients paradoxically develop a potentially lethal prothrombotic immune-complex-mediated reaction, characterized by moderate to severe thrombocytopenia and high risk of thrombosis [1,2]. Due to the widespread use of heparin, HIT is considered one of the most common adverse drug reactions, and the most frequent form of immune-mediated drug-induced thrombocytopenia. If unrecognized, HIT may be associated with significant morbidity and mortality. Unfortunately, since thrombocytopenia is very common in various medical conditions, a diagnosis of HIT may be missed. For patients receiving unfractionated heparin (UFH), the frequency of HIT can reach 3% in postoperative setting and 1% for other medical situations [2,3,4]. The immune-mediated mechanism that characterizes HIT represents an atypical immune response because it has features of both T-cell-dependent and T-cell-independent mechanisms. The disorder is characterized by newly formed immunoglobulin G (IgG) antibodies, which are characteristic of a T-cell-dependent mechanism. However, re-exposure to heparin, months after HIT, does not lead to a memory response, which is consistent with a T-cell-independent mechanism. The pathogenic IgG antibodies produced by HIT patients are directed against complexes formed by platelet factor 4 (PF4) and heparin (H) [5,6,7]. PF4 is a tetrameric protein released by α granules following platelet activation. At stochiometric concentrations, PF4 and heparin form complexes (PF4/H) through electrostatic interactions that trigger the generation of anti-PF4/H IgG antibodies. Immune complexes formed by PF4/H and anti-PF4/H IgG crosslink Fc-receptors through the binding of the IgG Fc region to FcγRIIA receptors, resulting in platelet and monocyte activation [8]. These events lead to the release of procoagulant chemokines and tissue factor, and of microparticles and platelet granules, which together create an intensely prothrombotic state associating platelet aggregation, formation of procoagulant platelets, and the activation of the clotting cascade leading to thrombin generation [9]. As a result, the activation of platelets induced by the binding of HIT antibodies promotes phenotypic changes at the surface of platelets, with the exposition of markers such as P-selectin and a change of the membrane polarity.

The first warning sign of HIT is the platelet count falling to 100 G/L or below, or a decreased platelet count by at least 30% between two successive counts. Hence, platelet count for early detection of HIT is systematically recommended. Platelet count monitoring is recommended 2–3 times a week for patients with high risk of HIT, and American society of hematology (ASH) guidelines suggest every 48 h for cardiac surgery patients with previous history of HIT [10,11]. In practice, the monitoring may even be performed on a daily basis in some worrying clinical situations. The risk of HIT is influenced by the type of heparin administered (UFH or LMWH), the underlying conditions, and the duration of treatment [10]. Indeed, UFH treatment is associated with higher risks than LMWH, which, in most cases, does not require any platelet count except in some situations such as cancer, patients with severe trauma, or postoperative care. Heparin is immediately discontinued when there is a suspicion, and an alternative anticoagulant must be used. As heparin is superior in many circumstances, it can be reintroduced only when HIT is ruled out. In order to help clinicians to manage patients with HIT suspicion, Lo et al. established a 4Ts score to calculate the pretest clinical probability of HIT prior to performing any laboratory test [2,12]. This calculation is based on the degree of thrombocytopenia, the timing of platelet count fall, thrombosis or other sequelae, and the likelihood of other causes for thrombocytopenia. The 4Ts rating scale ranges from 1 to 8, and in case of an intermediate (4–5) or high (6–8) score, a laboratory diagnostic test must be performed. Currently there is no ideal laboratory test for HIT [13], and it is recommended that a diagnostic algorithm for HIT diagnosis work-up is followed. Immunological assays such as ELISAs, particle gel immunoassays, and particle immunofiltration assays detect the binding of antibodies to PF4/polyanion complexes in patients’ blood [1]. Negative results from highly sensitive IgG-specific immunoassays rule out HIT, and no further laboratory investigation is needed. However, these immunoassays are not able to discriminate pathogenic from non-pathogenic HIT antibodies. Therefore, performing a functional assay is required to confirm the diagnosis of HIT in the presence of positive immunoassay results. All functional assays share the basic principle of demonstrating that donor platelets incubated with patient serum or plasma and a therapeutic concentration of heparin (0.1–0.5 IU/mL) are activated, while there is no significant activation with high concentration of heparin (100 IU/mL). Though regarded as gold standards for the diagnosis of HIT, serotonin release assay (SRA) and heparin-induced platelet activation (HIPA) are technically demanding, time-consuming, and are only performed every few days or even weeks in few remote and highly specialized laboratories (with SRA using radioactivity) [14]. In addition, they lack a proper external quality assessment scheme [15]. Consequently, although HIT suspicion is an urgent clinical concern, results are rarely available in a timely manner [16], which leads to the risk of unduly discontinuing heparin and exposing patients to bleeding risk associated with novel anticoagulant. Several other functional assays have been developed, mostly inhouse, to overcome these limitations [13]. These assays differ in the platelet activation endpoint measured, the technology and methodology used, the platelet donor selection, the platelet suspension (washed platelets, platelet-rich plasma, or whole blood), the patient sample (serum or plasma), and the heparin used (type and concentrations). Unfortunately, the lack of standardization of these assays limits the evaluation and the accessibility of functional assays in laboratories. There is thus a need for novel, less technically demanding laboratory assays that provide results rapidly.

Alternative methods of HIT diagnosis based on flow cytometry have emerged over the past years [13]. The idea of using flow cytometry in HIT diagnosis is not new, but it has become a promising method of confirming HIT thanks to recent advances of the technology. The advantages of flow cytometry over other functional assays in the diagnosis of HIT are their non-radioactivity, objective analysis, rapid turnaround time, patient samples in the microliter range, and no need for additional laboratory equipment. In addition, the single-cell-based multiparametric dimensions of flow cytometry measurement provides information-rich results that contrast with averaged measurements obtained over an undifferentiated platelet population by conventional bulk assays, such as SRA, HIPA, or platelet aggregometry. Flow cytometers are now widespread in clinical laboratories. From the early technical development in the 1950s to today, flow cytometry has become indispensable in hematology and immunology in the routine diagnosis of blood cancers such as leukemia and lymphoma and immunodeficiencies such as HIV infection [17]. It has also led to great advances in many biomedical areas such as molecular biology, bacteriology, virology, cancer biology, and infectious disease monitoring [18,19,20]. Moreover, progress in the technology has democratized flow cytometry to the benefit of its availability. Flow cytometry can be used to obtain multiparametric information on thousands of cells in a fluid stream passing through a light source in seconds by measuring the optical and fluorescence characteristics of each individual cell [13]. Size, granularity, and fluorescent features of the cells, derived from either antibodies or dyes, are among the parameters used to analyze and differentiate the cells. The underlying principle of flow cytometry is related to light scattering and fluorescence emission, which occurs as light from a laser beam strikes the moving cells. The data obtained give valuable information about functional status of cells, such as the degree of activation of platelets. From the perspective of quantifying platelet activation induced by heparin, it is important to note that fluorescence emission derived from a fluorescence probe is proportional to the number of fluorescent probes bound to the cell or cellular component [18]. A variety of flow cytometry functional assays, mostly inhouse protocols, are available for HIT diagnosis [13]. In these tests, platelets in suspension are typically labeled with fluorescent conjugated monoclonal antibodies as fluorescent markers before analyzing the sample. Researchers systematically used markers against specific platelet glycoproteins such as CD41, CD42, and CD61 to identify platelet population [21], but they often differ in the use of platelet activation markers such as annexin V [22,23], CD62p [24,25,26], and the detection of platelet-derived microparticles [27,28]. Alternatively, Gobbi et al. proposed two unconventional methods. The first one is a non-radioactive method for the detection of serotonin content using specific anti-serotonin intracellular staining [29]. The second method uses PF4/heparin-coated beads for platelet determination [30]. Despite the emergence of these new flow-cytometry-based methods, there is still a lack of standardization between all these different protocols.

In the present article, we describe the method development and optimization of a novel flow-cytometry-based rapid functional assay (HIT-FCA) to help establish or confirm HIT diagnosis. This assay uses two fluorescent antibodies, anti-CD41 and anti-CD62p, as markers. As with any HIT functional assay, HIT-FCA aims at demonstrating that donor platelets incubated with patient serum or plasma and therapeutic concentration of heparin (0.1–0.5 IU/mL) are activated, while there is no significant activation with high concentrations of heparin (10 to 100 IU/mL). HIT-FCA was adapted from A. Tomer’s seminal assay methodology [31,32], which subsequently led to the development of a commercialized test in compliance with ISO 13485:2016 norms and the European regulatory context (HIT Confirm^®^, Emosis, France). We focus this article on the description of the analytical development from the initial assay to a standardized test, which was achieved by working on the optimization of the fluorescent signal, the platelet-rich plasma (PRP) selection, the evaluation of the repeatability and reproducibility, and the establishment of a heparin-induced platelet activation index allowing a straightforward intra-laboratory comparison.

## 2. Materials and Methods

### 2.1. Materials

Two clones of phycoerythrin (PE)-conjugated anti-CD41, monoclonal antibodies (MoAbs) directed against platelet glycoprotein IIb/IIIa, were purchased from Dako Glostrup, Danemark (clone 5b12) and Becton Dickinson, Franklin Lakes, NJ, USA (clone HIP8). Fluorescein isothiocyanate (FITC)-conjugated anti-CD41 monoclonal antibodies were purchased form Merck (clone 5b12) and Becton Dickinson, Franklin Lakes, NJ, USA (clone HIP8). FITC- and PE-conjugated anti-CD62p monoclonal antibodies, recognizing activated platelet P-selectin, were purchased from Becton Dickinson, Franklin Lakes, NJ, USA (clones AC1.2 and AK4) and BioRad, Hercules, CA, USA (clone AK6). The various thrombin-receptor-activating peptides (TRAPs) tested, named TRAP-1, TRAP-2, and TRAP-3, were purchased from Merck, Darmstadt, Germany (MW 1740, 1 mg/mL), Phoenix Pharmaceuticals, Inc., Burlingame, CA, USA (MW 747, 1 mg/mL), and Biotechne, Minneapolis, MN, USA (MW 1739, 1 mg/mL), respectively. Dilution buffer, Dubelcco’s phosphate buffered saline 10× w/o calcium w/o magnesium (from Dominique Dutscher, Brumath, France) was diluted 10-fold before use (PBS 1×). Three different heparins were supplied by Sigma-Aldrich, Saint Louis, MO, USA: heparin sodium salt from porcine intestinal mucosa (Grade I-A, ≥180 United States Pharmacopeia USP) units/mg, Heparin-1), heparin sodium salt from porcine intestinal mucosa (≥150 UI/mg, Heparin-2), and heparin sodium porcine mucosa suitable for use with in vitro diagnostic kits, which meets USP testing specifications (Heparin-3).

### 2.2. Preparation of Platelet-Rich Plasma and Platelet-Poor Plasma

Whole blood from unmedicated healthy volunteers from a blood bank was drawn into sodium citrate vacuum tubes (0.109 M), rested for 30 min before use, and centrifuged at 200× *g* for 5 min at room temperature with no brakes. The platelet-rich plasma (PRP) was collected in a new tube and was immediately used for tests. Citrate vacuum tubes without PRP were centrifugated at 2000× *g* for 10 min at room temperature with no brakes. The platelet-poor plasma (PPP) was collected in a new tube and was immediately used for tests.

### 2.3. Samples

For the analytical development, samples were purchased from a sample provider (Clinisys Associates Ltd., Atlanta, GA, USA). Samples were characterized as HIT-positive sera by ELISA and SRA or HIT-negative (normal donor plasma) by the supplier. Samples were qualified internally by Emosis (Illkirch, France) as high control (QCH), medium control (QCM), or blank control (QCB) using ELISA, SRA results, and activation percentage using HIT-FCA.

### 2.4. Anti-CD41 Titrations

A range of 0 to 4 µL of PE- or FITC-conjugated anti-CD41 was incubated with 5 µL PRP, 10 µL of PPP, and PBS 1× (final volume, 50 µL) for 20 min. The samples were finally diluted with 450 µL of PBS 1× and immediately read by flow cytometry.

### 2.5. Anti-CD62 Titrations

A range of 0 to 8 µL of PE- or FITC-conjugated anti-CD62p was incubated with 1.25 µL of PRP, 5 µL of PPP, 4 µL of TRAPs, and PBS 1× (final volume, 25 µL) for 20 min. The samples were finally diluted with 250 µL of PBS 1× and immediately read by flow cytometry.

### 2.6. TRAP Titrations

A range of 0 to 8 µL of TRAPs was incubated with 1.25 µL of PRP, 5 µL of PPP, 0.5 µL of PE-conjugated anti-CD41, 1 µL of FITC-conjugated anti-CD62p and PBS 1× (final volume, 25 µL) for 20 min. The samples were finally diluted with 250 µL of PBS 1× and immediately read by flow cytometry.

### 2.7. Heparin Titrations

The patient sample (10 µL) was incubated with PRP (10 µL) and a range of 0 to 100 IU/mL heparin for 1 h at room temperature (final volume, 50 µL). Following the first incubation, PE-conjugated anti-CD41 and FITC-conjugated anti-CD62p were added (final volume, 25 µL) and incubated for 20 min at room temperature. The samples were finally diluted with 250 µL of PBS 1× and read by flow cytometry.

In parallel, as positive control (POS) for platelet activation, 2.5 µL donor PRP with 2.5 µL donor PPP was treated with 2 µL TRAPs and PE-conjugated anti-CD41 and FITC-conjugated anti-CD62 for 20 min (final volume, 25 µL). A negative control (NEG) was also prepared the same way without TRAPs.

### 2.8. Reference Values and PRP Stability

The normal or patient sample (10 µL) was incubated with unmedicated healthy donor PRP (10 µL) and with either low-dose heparin (0.3 IU/mL) or high-dose heparin (100 IU/mL) for 1 h at room temperature (final volume, 50 µL) with PE-conjugated anti-CD41 and FITC-conjugated anti-CD62. In parallel, positive and negative controls were prepared. The samples were finally diluted with 450 µL of PBS 1× and immediately read by flow cytometry.

### 2.9. Repeatability and Reproducibility Analysis

The patient sample (10 µL) was incubated with PRP (10 µL) and 0.3 IU/mL or 100 IU/mL heparin for 1 h at room temperature (final volume, 50 µL). Following the first incubation, PE-conjugated anti-CD41 and FITC-conjugated anti-CD62p were added (final volume, 25 µL) and incubated for 20 min at room temperature. The samples were finally diluted with 450 µL of PBS 1× and immediately analyzed by flow cytometry. In parallel, as positive control (POS) for platelet activation, 2.5 µL donor PRP with 2.5 µL donor PPP was treated with 2 µL TRAPs and PE-conjugated anti-CD41 and FITC-conjugated anti-CD62 for 20 min (final volume, 25 µL). A negative control (NEG) was also prepared the same way without TRAPs. For repeatability and reproducibility, three levels of control samples were selected and internally qualified: QCH as high control (heparin-induced platelet activation index, %HEPLA, ≥ 50%), QCM as medium control (13% < %HEPLA < 50%), and QCB as blank control (%HEPLA < 9.6%). QCH and QCM were beforehand characterized as HIT-positive sera by SRA.

### 2.10. Flow Cytometric Acquisition

Acquisition of flow cytometry data was performed by using a BD FACSVia flow cytometer (Becton Dickinson, San Jose, CA, USA) equipped with blue and red lasers, two light scattering detectors and four fluorescence filters. For the assay, both fluorophores were excited by the blue laser at 488 nm. FL1 (533/30 nm) and FL2 (585/40 nm) filters were used to detect fluorescence emission of FITC and PE fluorophores, respectively. Daily internal quality control of the instrument was performed using BD CS&T beads (Becton Dickinson) according to the manufacturer’s protocol. Fluorescence compensation was applied to correct emission spectra overlap between FITC and PE fluorophores using single-color-labeled preparations. Ten thousand events in the CD41+ gate were acquired for each sample. Different features were recorded for each event such as size, granulometry, and fluorescence intensity in FL1 and FL2 channels.

### 2.11. Flow Cytometry Analysis

Flow cytometry data were analyzed with BD software (Becton Dickinson). First, the platelet population was displayed on a density dot plot of log FSC vs. log SSC to ensure that the population was unique. Then, platelets were discriminated from debris by fluorescence emission in the FL2 channel corresponding to the PE signal. Platelet population (CD41+) was gated on a single-parameter FL2 histogram. Except for anti-CD41 titrations, a marker indicating activation threshold was placed at the intersection of single-parameter FL1 (FITC) histograms for POS and NEG. This analytical cutoff defines the gate of CD62+ platelets corresponding to the fraction of activated platelets (activation %). Graphically, it is the percentage of platelets with a fluorescence intensity beyond this threshold. Then, heparin-induced platelet activation index (% HEPLA) was calculated as follows:(1)$$\%\mathrm{HEPLA}\;=\;\;\frac{\%H0.3-\%H100}{\%\mathrm{POS}-\%\mathrm{NEG}}$$
where %NEG, %POS, %H0.3, and %H100 are fractions of activated platelets for NEG, POS, H0.3, and H100 tubes, respectively.

Obtained data were then compiled with GraphPad Prism Software (San Diego, CA, USA) and Microsoft Excel (Microsoft Corporation, version 2102).

## 3. Results

### 3.1. Principle of HIT-FCA

HIT-FCA is a flow-cytometry-based HIT functional assay, aimed at assisting clinicians in the diagnosis or confirmation of HIT along with 4Ts score and immunoassays. Unlike usual HIT functional assays, this test is rapid and simple to perform routinely in any laboratory equipped with a flow cytometer. Therefore, HIT-FCA is intended to be used in acute situations as well as for monitoring HIT evolution. The principle of the assay is shown Figure 1. Plasma from a suspected HIT patient is tested with healthy donor PRP. Patient serum and PRP are separately incubated 60 min with a low heparin concentration of 0.3 IU/mL (H0.3) and a high heparin concentration at 100 IU/mL (H100). The first heparin concentration corresponds to the range in which PF4/H complexes are formed leading to platelet activation. However, at high concentrations of heparin, platelet activation does not occur. In parallel, negative (NEG) and positive (POS) controls comprising only donor platelets and donor platelets with thrombin-receptor-activating peptides (TRAPs), a platelet activator, respectively, are used. Then, two surface markers, fluorochrome-conjugated anti-CD41 and anti-CD62p MoAbs, are added to the reaction followed by 20 min incubation before reading by flow cytometry. These fluorescent-labeled antibodies are used to quantify the platelet activation level. The first one, anti-CD41, recognizes CD41 receptors on both resting and activated platelets. This helps us to identify and to gate the platelet population excluding all the other cells or debris potentially found in PRP or patient serum. The second one, anti-CD62p, is directed against P-selectin receptors, which are expressed at the surface of activated platelets. Accordingly, the detection of both markers’ fluorescence characterizes the activation of platelets. To the contrary, the detection of the sole fluorescence of the fluorochrome-conjugated anti-CD41 is indicative of non-activated platelets. Finally, the assay takes less than 2 h to be completed once PRP is ready.

### 3.2. Optimization of Reagent Concentrations

For optimization of HIT Confirm, titrations of the different reagents were performed (Figure 2). Two clones of anti-CD41 were tested both coupled with either FITC or PE fluorochrome (Figure 2a). Maximal median fluorescence intensity (MFI) was obtained with PE-conjugated 5B12 clone, and saturation was obtained with 2 µL of this antibody. In parallel, anti-CD62p was tested (Figure 2b). Maximal MFI was obtained with PE fluorochrome; indeed, this fluorochrome is brighter than FITC. PE-conjugated anti-CD41 gave better results, and another fluorochrome for anti-CD62 was selected. The three FITC-conjugated anti-CD62 clones showed the same dose–response curve. Maximal MFI was obtained with the clone AK6, and saturation was obtained with 0.125 µL of this antibody. For TRAPs, two distinct patterns were obtained (Figure 2c). TRAP-2 reached saturation with lower volume than the two other TRAPs. Volume of 4 µL is necessary with TRAP-2, and the maximal activation was obtained with this volume. For antibodies and TRAPs, the combination of maximal MFI and saturation volume was used to select the reagents. A final verification of the supplier price was conducted internally if same results were obtained. To avoid excessive background noise, the reagent volumes selected for the assay protocol were slightly below saturation values. For heparin, two concentrations were required to realize the test: a therapeutic concentration that activates platelets in the presence of HIT antibodies and an excessive concentration that prevents the formation of PF4-heparin complexes and thus inhibits the anti-PF4/H IgG-mediated activation of platelets (Figure 2d). Maximum activation was obtained with 3 IU/mL for all three heparins. No activation was obtained with 100 IU/mL. As heparin-3 was more cost-effective, its supplier was selected.

### 3.3. Reference Values and PRP Stability over Time

The stability of donor platelets was investigated by measuring the % activation of negative and positive controls (Figure 3). The percentage of activation corresponds to the proportion of platelets beyond the cutoff delimiting the NEG and POS curves in the FL1-signal. First, 25 batches of healthy donor PRPs were run with or without TRAPs. The % activation obtained is represented in Figure 3a,b. Means (solid lines) and ± 2 standard deviations (± 2 SDs, dashed lines) were then calculated. According to these values, the high limit for negative control was set at 19.3% activation and the low limit for positive control at 84.9% activation. This means that outside of these limits, PRP would not be suitable for the assay. In addition, an activation < 2.9% of negative control indicated poorly reactive donor platelets. Hence, we recommend using only PRP with a % activation of NEG and POS included in these reference values. The stability of donor platelets over time was tested with seven PRP batches (Figure 3c,d). These batches were tested several times between 60 and 360 min following PRP preparation. For the majority of PRPs, the percentages of activation were included in the reference values and therefore were considered as stable even after 360 min. However, for three batches, the % activation was out of the thresholds of 19.3% activation for negative control or 84.9% activation for positive control over 160 min. For this reason, we considered that the assay must be performed with fresh PRP not older than 3 h after PRP preparation.

### 3.4. Repeatability and Reproducibility

The precision of the obtained measures was determined by the repeatability and reproducibility of the assay. For this purpose, we used three types of samples, HIT-confirmed sera (QCM and QCH) and normal serum (QCB). The study was carried out with platelets obtained from different donors, and %HEPLA was calculated for each control sample. Repeatability was assessed by a single operator, on the same day, with six repetitions for each sample. Alternatively, reproducibility was carried out by several operators, for nine independent experiments and different days. Means of %HEPLA, standard deviation (SD), and coefficient of variation (%CV) were calculated (Table 1). QCM and QCH were systematically qualified as HIT-positive (i.e., %HEPLA > 13%) with respect to reference values previously defined for NEG and POS. In addition, their CVs were lower than 10% for repeatability and reproducibility except the reproducibility of QCM, for which CV reached 23.6%. Even if a %CV of 10% is desired, up to a 25 %CV is acceptable for QCM and QCH samples [33]. However, it is essential that results of QCB samples are always negative to ensure the specificity of the assay. For QCB, 100% of the results qualified the samples as HIT-negative (i.e., %HEPLA < 9.6%). Nevertheless, %CV of QCB was higher than 25% for both analyses. We assumed that high %CV expresses the variability of donors.

## 4. Discussion

The HIT-FCA assay involves a two-color flow cytometry protocol including two successive stages, reduced to one step in the commercialized version, applied to patient plasma samples, preferably to serum samples, using non-washed platelets. While serum can be used, plasma offers distinct advantages over serum with respect to standardizing the HIT functional assay. In plasma, interference due to coagulation is eliminated, as coagulation post-centrifugation does not occur. Interference with coagulation in serum leads to the lack of some coagulation factors that are consumed during the clotting and includes latent fibrin formation when clotting is inadequate or in samples from patients receiving anticoagulant or thrombolytic therapy. In addition, interference due to coagulation in serum includes complex reciprocal coagulation–complement interactions, including activation and aggregation of platelets, which in turn contribute to the amplification of complement. This coagulation–complement interaction justifies heat inactivation of serum complement. However, though complement is not activated in normal plasma, it can be activated in certain circumstances such as cardiovascular surgery [34]. Other advantages of plasma over serum are a lower risk of hemolysis and thrombocytosis and, most significantly, the in vivo state of constituents in plasma remains unchanged after sampling, which makes plasma more accurately representative of the in vivo status of the patient than serum [35]. However, new assays using plasma should be tested for anticoagulant interference, since anticoagulants can—as potential complexing agents and enzyme inhibitors—lead to method-dependent interference [35]. In addition, from a practical point of view, since clotting—the time range of which is from 30 to 45 min, or even longer for patients on anticoagulant therapy—is not required for plasma, it enables plasma to be centrifuged upon receipt of the specimen in the laboratory and to be processed more quickly, shortening the turnaround time for test results. In addition, there is a potentially higher sample volume yield with plasma, with approximately 15–20% more plasma obtainable from whole blood than with serum. Incidentally, this helps laboratories to adhere to ISO standard 15189, according to which laboratories should periodically review sample volume requirements to ensure that excessive quantities of blood samples are not collected.

Flow cytometry analysis typically begins with the critical step of distinguishing cells of interest and of placing gates around the regions containing the populations of those cells. Accordingly, the first stage of the assay is to detect platelets from other blood cells according to their small size and granulometry and to confirm this identification using anti-CD41-PE to exclude other cells or cell debris for the analysis (Figure 1 and Figure 4). Anti-CD41-PE is a fluorescent-conjugated MoAb labeling platelets exclusively through its specific binding to the platelet membrane glycoprotein CD41, independently of their state of activation. To quantify the fraction of the platelet population that is activated in suspected HIT patients, activated platelets are detected with another fluorescent-conjugated MoAb, anti-CD62-FITC, that specifically binds the membrane glycoprotein P-selectin (also known as CD62 or CD62p). At rest, P-selectin receptor is localized on the inner walls of α-granules. Platelet activation triggers α-granule release and the translocation of P-selectin to the platelet surface [13]. Fluorescent-conjugated annexin V, a positively charged protein, may also be used to detect activated platelets in flow cytometry. However, Vitale et al. showed that P-selectin expression was positive in 85% of HIT cases compared to only 40% for annexin V [36]. For this reason, anti-CD62-FITC was chosen as the best marker of activated platelets for the diagnosis of HIT. The principle of HIT-FCA is conceptually close to the principle of the gold standard SRA. In SRA, the platelet activation marker is the release of serotonin from platelet dense granules, measured using ^14^C radiolabeled serotonin, and expressed as a percentage. As shown in Figure 2a,b, different clones of anti-CD41-PE and anti- CD62-FITC were compared and titrated. This key step of analytical development enabled us to define the optimal concentration of both fluorescent-conjugated MoAbs with respect to assay resolution and choose the best clones to source these MoAbs, for the sake of assay reliability and analytical accuracy.

As previously stated, the use of two concentrations of heparin, H0.3 and H100, as seen in Figure 1, is essential in HIT-FCA. Indeed, the stoichiometry of PF4 and heparin is a key issue in the mechanism of HIT. Rauova et al. demonstrated that PF4 and UFH form ultra-large complexes (ULCs) over a narrow ratio of molar concentrations [37]. These observations were correlated with the titration of heparin in our test (Figure 2d). Based on these results, we assumed that stable ULCs are formed over a range of low heparin concentrations (0.1–1 IU/mL), which could be targeted by HIT antibodies eventually promoting platelet activation. However, over a range of high heparin concentrations (10–100 IU/mL), HIT antibodies are unable to activate platelets since the broken stoichiometry balance inhibits the formation of ULCs. Such a pattern of heparin-dependent platelet activation enables us to exclude other causes of antibody-mediated platelet activation, resulting in a higher specificity as compared to immunoassays [38]. Accordingly, 0.3 (H0.3) and 100 IU/mL (H100) were selected as the low and the high heparin concentrations, respectively. These concentrations are consistent with the ranges of heparin concentrations used for other functional tests [39]. In the analysis, NEG and POS were used to set the cutoff to separate activated platelets from non-activated platelets in the FL1 signal (Figure 4). In the initial methodology, analytical cutoff was defined as the NEG’s mean fluorescence intensity + 2 SDs [31,32]. However, this approach has limits. The distribution of measured events over the fluorescence intensity expressed in log is assumed to be normal, which may not be always the case. In addition, the threshold value, which must be computed for each PRP prepared from platelets supplied by a different donor, is often difficult to position precisely on the log axis of number of events. Moreover, setting the threshold from the fluorescence baseline mean favors limiting the rate of false negatives (e.g., to 5% in the case of 2 SDs, assuming a normal distribution), hence the analytical sensitivity of the assay. The novel approach of setting the threshold at the intersect between NEG and POS offers an easier and more objective way of setting the analytical cutoff (Figure 4). To minimize the intrinsic variability of activation from donor platelets, we standardized results with %HEPLA, which expresses the percentage of platelet activation induced by heparin as a proportion of donor platelet activation potential. The numerator of %HEPLA, Equation (1), reflects the expectation that a therapeutic concentration of heparin triggers platelet activation, while a high concentration of heparin does not. Hence, pathological antibodies induce an increase of %H0.3 minus %H100. The denominator of %HEPLA reflects the width of the activation potential of donor platelets. It is calculated as %POS minus %NEG, with %NEG being the platelets basal unstimulated state and %POS being a strong agonist-induced extremum of platelet activation. However, in some circumstances, as reported with SRA, platelet activation is triggered by a low concentration of heparin, but it is not inhibited by a high concentration of heparin, with high concentration of heparin still triggering a serotonin release above 20% [40]. In such cases, SRA cannot formally exclude HIT, and repeating the test leads to interpretable results in about half of the cases. Similarly, %H100 remaining high may occur with HIT-FCA, which may result in a low %HEPLA because of the shrinking of the difference between %H0.3 and %H100. As with SRA, the lack of inhibition of platelet activation by high concentration of heparin leads to indeterminate, non-conclusive results. Accordingly, to avoid low or even negative %HEPLA associated with elevated %H100 being falsely interpreted as negative, it is important to follow the rule of examining %H100 when %HEPLA is low (Figure 5). A low %HEPLA associated with an elevated %H100 is thus indeterminate. This must lead to repetition of the test with other PRPs, as suggested by Moore et al. for SRA [40]. If the repeated results are still indeterminate, the HIT-FCA result is deemed inconclusive. A precise cutoff for “high” %H100 is somewhat elusive. However, since a mean ± SD value of %NEG of 11.1% ± 6.4% was measured in a large cohort of HIT patients [41], we propose to define %H100 as being high when it is above 23%.

A circumstance that may falsely lead to low or even negative %HEPLA, with the test result being possibly misinterpreted as negative accordingly, is delaying the flow cytometric measurement once the sample is prepared. The use of platelets makes the assay time-sensitive, and it is thus imperative to perform the assay without delay once the sample is ready for testing. A critical issue in HIT functional tests is precisely the platelet donor selection because of the high inter-variability of platelet donors [42]. In the present article, we suggest some tips to limit this PRP variability. First, blood donors must be free from non-steroidal anti-inflammatory drugs and aspirin, which are known to affect platelet function for at least 7 days prior to blood collection. Besides the influence of some medications or diets, the variability of platelets might be explained by the polymorphism of FcγRIIA receptors [43] or the variation of PF4 concentrations [44] on the platelet surface. Therefore, it is recommended to use plasma (or serum) from a patient with confirmed HIT as the positive control to ensure the platelet reactivity before performing the test on suspected HIT patient plasma. If such positive HIT plasmas (or sera) or donors whose platelets are known to be reactive to anti-PF4/H antibodies are not available, it is key to multiply PRPs from different unselected donors. Accordingly, we recommend performing the assay using at least two different PRPs and applying the “believe the positive” decision rule. Assuming that the polymorphism of the FcγRIIA receptor is the main cause for the lack of platelet activation in the presence of pathogenic anti-PF4/H antibodies, and since the frequency of the low affinity variant genotype of the FcγRIIA receptors is about 25% on average [45], the probability of having a low responder PRP from one unselected platelet donor is 25%, 6.25% with two, and 1.5% with three. Whether using PRPs known to be reactive when exposed to HIT+ plasma (or serum) or unselected PRPs, the reference values set previously and represented in Figure 3a,b help in excluding insufficiently responsive PRP (i.e., %NEG < 2.9% of activation or %POS < 84.9% of activation). Likewise, it is possible to exclude over-reactive PRPs (i.e., %NEG > 19.3% of activation), which is often due to excessive handling. Results in Figure 3c,d also suggest a time-dependent variability of reactivity for some PRPs as stated above.

Recently, it was suggested to select platelet donors or to validate PRPs by pre-testing responsiveness to anti-PF4/H antibodies using engineered antibodies mimicking the effect of pathological anti-PF4/H, such as the clone KKO, a mouse monoclonal antibody previously shown to bind PF4/H complexes or 5B9, a chimeric monoclonal anti-PF4/H antibody with a human Fc fragment [46]. The latter was recently tested on the latest version of HIT Confirm^®^ in a study comparing different current methods [47]. On another note, a supplementation of PF4 showed interesting results by improving the sensitivity of some functional assays [48,49].

The present assay, HIT-FCA, was evaluated in a retrospective study involving 650 samples and compared to a particle gel immunoassay (PaGIA). HIT-FCA showed significantly higher correlation with 4Ts compared with the PaGIA method [50]. In parallel, our commercialized CE-marked assay (HIT Confirm^®^, Emosis, France) was also compared with such other conventional functional tests as HIPA [51], SRA, and heparin-induced multiple electrode aggregometry (HIMEA) [47].

## 5. Conclusions

Flow cytometry is a potentially widely available alternative to functional tests currently performed in a restricted number of specialized laboratories. HIT-FCA is a rapid functional assay with satisfying reproducibility and repeatability, which enables a timely functional testing of the pathogeny of anti-PF4/H antibodies when a presumed diagnosis of HIT necessitates an urgent workup. HIT-FCA is a first-generation flow-cytometry-based assay, the analytical development of which described above paved the way for a standardized commercialized assay with a diagnostic performance that is assessed retrospectively and is to be further validated by a prospective study. Opportunities exist to develop a flow-cytometry-based second generation of HIT functional assay, benefiting from improvements of quality control of platelets, addition of performance enhancing reagents, and automated gating and algorithmizing data analysis.

## Figures and Tables

**Figure 1 biomedicines-09-00296-f001:**
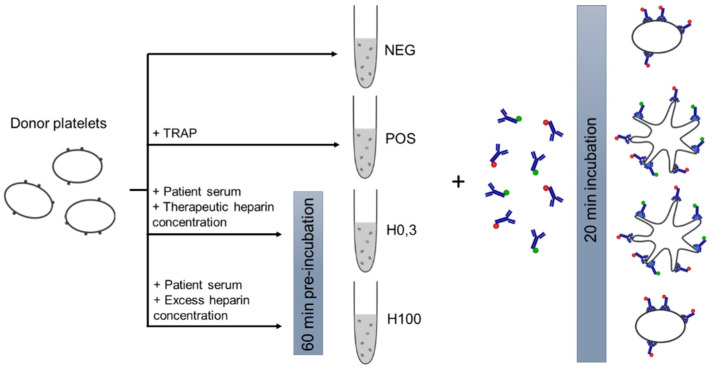
Principle of HIT flow cytometry assay (HIT-FCA), a flow-cytometry-based HIT functional assay, using donor platelets and dye-conjugated monoclonal antibodies. This assay uses four tubes: a negative control (NEG), a positive control (POS), and two tubes with patient sample containing different heparin concentrations (H0.3 and H100). The PE-conjugated anti-CD41 antibody (red) specifically labels platelets. In the presence of anti-PF4/H antibodies, donor platelets are activated, and they express CD62 or P-selectin in their plasma membrane. The latter is then recognized by the second FITC-conjugated anti-CD62 antibody (green). The tubes are read by flow cytometry after the incubation times.

**Figure 2 biomedicines-09-00296-f002:**
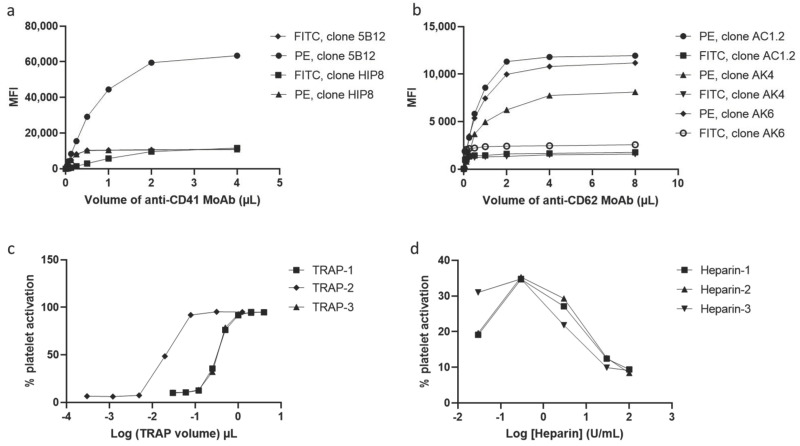
Optimization of assay reagents. The median fluorescence intensities (MFIs) of (**a**) two anti-CD41 and (**b**) three anti-CD62 clones were measured. These monoclonal antibodies (MoAbs) were labeled with either FITC or PE fluorophores. Three different suppliers for (**c**) thrombin-receptor-activating peptides (TRAPs) and (**d**) heparin were also tested.

**Figure 3 biomedicines-09-00296-f003:**
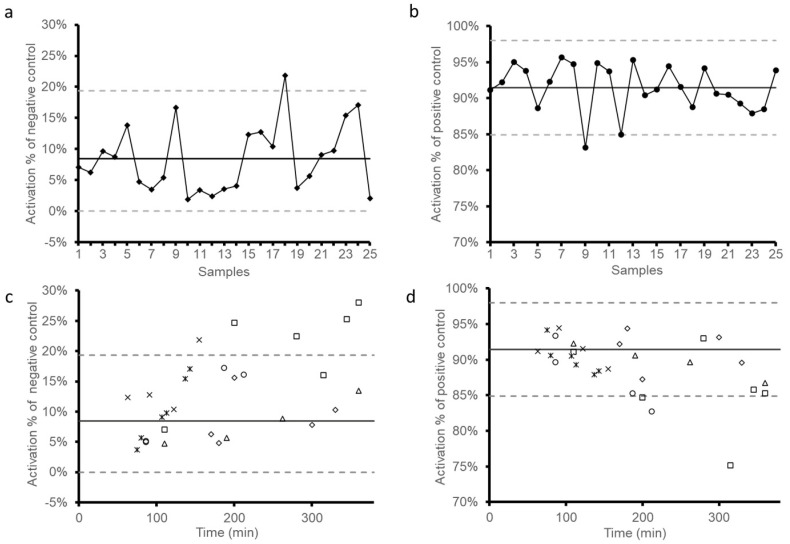
Donor platelet stability. (**a**) Negative and (**b**) positive reference values for HIT Confirm were set according to 25 batch samples of donor platelets. The solid lines correspond to the mean, and dashed lines show the limits of mean ± 2 standard deviations (SDs). The change of activation percentage of (**c**) negative and (**d**) positive controls as a function of time overlayed with the mean and limits of mean ± 2 SD as defined in the previous plots. Seven batch samples were tested.

**Figure 4 biomedicines-09-00296-f004:**
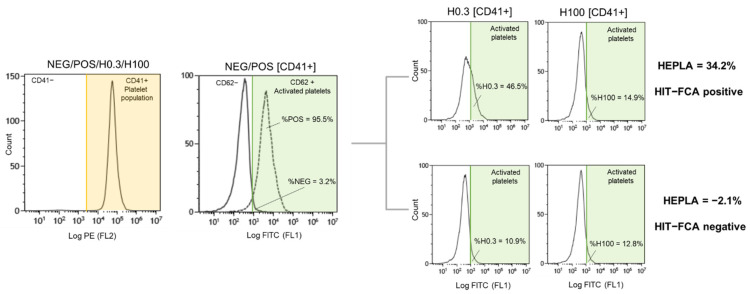
Gating strategy for HIT-FCA assay. Left: PE profile in the four tubes used to identify platelet population. Middle: negative (NEG) and positive (POS) FITC profiles in solid and dashed lines, respectively, overlayed to define the analytical cutoff of activated platelets. Right: examples of HIT-FCA-positive and HIT-FCA-negative with the calculated %HEPLA.

**Figure 5 biomedicines-09-00296-f005:**
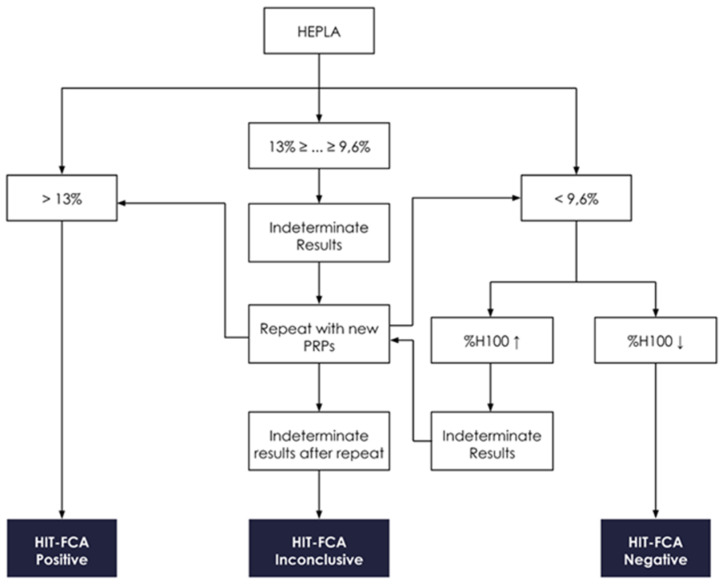
HEPLA interpretative algorithm. If HEPLA is above 13%, the HIT-FCA result is considered positive [41]. In case of HEPLA below 9.6%, %H100 is to be checked; if %H100 is high (↑)—that is, above 23% (see text)—the result is considered indeterminate, and the assay must be repeated with new PRPs. If %H100 is low (↓), the HIT-FCA result is considered negative. If HEPLA is in the gray zone from 9% to 13%, the HIT-FCA result is considered indeterminate. If indeterminacy of HEPLA remains after repeats, the HIT-FCA result is deemed inconclusive.

**Table 1 biomedicines-09-00296-t001:** Repeatability and reproducibility: mean of heparin-induced platelet activation index (%HEPLA), standard deviation (SD), and coefficient of variation (%CV).

	Repeatability			Reproducibility		
	Mean	SD	%CV	Mean	SD	%CV
QCH	92.6%	2.4%	2.6%	91.4%	5.0%	5.5%
QCM	26.5%	2.4%	9.1%	22.7%	5.4%	23.6%
QCB	1.2%	1.1%	88.0%	2.6%	2.6%	102.6%

QCH: high control; QCM: medium control; QCB: blank control.

## Data Availability

The data are not publicly available as they are part of the confidential regulatory technical documentation of a marketed product.

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
