# Peer review of "An Optimized and Standardized Rapid Flow Cytometry Functional Method for Heparin-Induced Thrombocytopenia"

_biomedicines, 2021, doi:10.3390/biomedicines9030296_

Round 1
Reviewer 1 Report
In this paper, Anne Runser et al., “describes and discusses the standardization of a HIT flow cytometry assay method (HIT-FCA) which subsequently led to the development and commercialization of a CE marked assay (HIT Confirm®, Emo-sis, France) as a suitable rapid HIT functional test.”.
The subject of the article are very interesting and relevant as Heparin-induced thrombocytopenia (HIT) is a complication of heparin therapy that could be a life-threatening side effect in HIT II, which is immunologically induced and often associated with thromboembolic complications. So, early detection of this type of HIT with rapid and standardized methods are needed.
However some considerations are need:
In introduction the authors need to clarify the two HIT types and perhaps the role of flow cytometry in the HIT detection.
The authors much refer how many samples are studied? How many volunteers and patients are included in the study?
The authors should refer to which heparin therapy the patients are submitted and what is the reason for this therapy? DVT? TEP? other? Is there any relation (interference) of these with the method described in the article.
Other comments:
When we use Flow cytometry for CD detection usually we need to exclude non-specific interaction using isotypes controls, but in the paper, I do not see any commentary on this. So, the authors need to clarify these aspect.
Reviewer 2 Report
Detailed experimental paper
please find my comments below
LMWH monitoring is required in some situations
frequency of platelet count monitoring (either daily or every two days) should be revised PMID: 32299756
the authors should summarize the literature about other flow cytometry assays used to diagnose HIT (pubmed 71 hits) and the added value of this new assay
citrate concentration: please use mM
please add an example of the flow cytometry results (raw data) (HIT+ and HIT-)
PPR --> PRP
the authors should provide a figure to explain how the cut-off was set
evaluation of this assay in real conditions should be provided. This review has found 3 papers
PMID: 31206215
PMID: 31997532
PMID: 31220752
Round 2
Reviewer 2 Report
thanks for addressing my comments